# An Innovative Analytical Approach for Multi-Mycotoxin Detection in Craft Beer Using Freeze-Dried Samples, IAC Column and HPLC/ESI-MS/MS

**DOI:** 10.3390/foods14060956

**Published:** 2025-03-11

**Authors:** Pietro Andronaco, Rosa Di Sanzo, Francesco Ioppolo, Francesco Ligato, Simone Alberto, Maria Angela Galluccio, Sonia Carabetta, Mariateresa Russo

**Affiliations:** Department of Agriculture Science, Food Chemistry, Safety and Sensoromic Laboratory (FoCuSS Lab), University of Reggio Calabria, Via dell’Università, 25, 89124 Reggio Calabria, Italy; pietro.andronaco@unirc.it (P.A.); francesco.ioppolo@unirc.it (F.I.); francesco.ligato@unirc.it (F.L.); simone.alberto@unirc.it (S.A.); maria.galluccio@unirc.it (M.A.G.); sonia.carabetta@unirc.it (S.C.); mariateresa.russo@unirc.it (M.R.)

**Keywords:** mycotoxins, craft beer, freeze-dried beer, HPLC/ESI-MS/MS, IAC column, food safety

## Abstract

The detection and quantification of mycotoxins in beer are critical for ensuring consumer safety and regulatory compliance. These contaminants, originating from barley and other grains, persist and potentially transform during the brewing process. This study presents an innovative analytical protocol using liquid chromatography/electrospray ionization tandem mass spectrometry (HPLC/ESI-MS/MS) for the simultaneous qualitative and quantitative analysis of nine mycotoxins, including aflatoxins (AFB1, AFB2, AFG1, AFG2), Ochratoxin A (OTA), Fumonisins (FB1, FB2), Deoxynivalenol (DON), and HT-2. The method leverages the efficiency of multi-mycotoxin immunoaffinity columns, providing streamlined sample preparation with high specificity and sensitivity. Validation was conducted using craft beers from Calabria, including freeze-dried samples to enhance analytical consistency and stability. The method’s accuracy was confirmed by using spiking samples with mycotoxins at concentrations compliant with the European Commission’s regulations (Recommendation 2024/1038/EU). The developed protocol delivers reliable results with minimized resource consumption, offering a robust tool for quality control and safety assessments in brewing. By addressing knowledge gaps in freeze-dried craft beer, this study contributes to advancing food safety standards in the brewing industry.

## 1. Introduction

Beer is the most widely consumed alcoholic beverage worldwide [1], but the presence of mycotoxins in beer raises significant public health concerns, particularly for those who consume it in large quantities [2]. The brewing process involves multiple steps that can influence the initial levels of mycotoxins, yet brewers lack absolute control over the chemical and biochemical reactions occurring in each batch [3]. The most effective strategy to prevent mycotoxin contamination remains the prevention of mold growth in raw materials.

Mycotoxins are secondary metabolites produced by filamentous fungi that have no biochemical role in fungal development but pose toxic risks to both animals and humans under favorable conditions [4]. The most critical mycotoxins from an agroeconomic and public health perspective include aflatoxins (AFs), ochratoxin A (OTA), patulin (PAT), trichothecenes (such as deoxynivalenol [DON], nivalenol [NIV], HT-2 toxin, and T-2 toxin), zearalenone (ZEN), fumonisins (FUM), tremorgenic toxins, and ergot alkaloids [3]. These mycotoxins are primarily produced by fungal genera such as *Aspergillus* (AFs, OTA, PAT), *Penicillium* (OTA, PAT), and *Fusarium* (DON, NIV, HT-2, T-2, ZEN) [5]. Their contamination extends across various food industries, affecting cereals, peanuts, dairy products, coffee, wine, beer, and other agricultural commodities [6].

Barley, along with water, hops, and yeast, is a fundamental ingredient in beer production, and its quality significantly influences beer’s overall acceptability [7]. Mycotoxin contamination can occur when barley, malt, hops, or other adjuncts are affected by fungal infection [8]. The most common fungal genera in malting barley include *Alternaria, Aspergillus, Penicillium,* and *Fusarium*, which can simultaneously produce multiple mycotoxins. Notably, nearly 30% of *Alternaria* fungi and 88% of *Fusarium* fungi isolated from barley grains have been found capable of producing various toxins such as *Alternaria* toxins, aflatoxins, OTA, DON, and ZEN [9].

Among mycotoxins in beer, DON has been the primary focus of research due to its prevalence and public health implications [3,10,11,12,13]. Since barley serves as the main ingredient, mycotoxins can enter the brewing process during malting, which involves soaking, germination, and kilning [14]. These steps create conditions where mycotoxins can persist, and in some cases, even concentrate. Thermal stability allows toxins like DON and FUM to survive the mashing and boiling stages, where barley malt is converted into fermentable sugars [15]. Fermentation, where yeast transforms sugars into alcohol, does not significantly reduce mycotoxin levels, as only small amounts are bound by yeast strains. Likewise, filtration and clarification steps do little to eliminate mycotoxins [15]. Even in packaging and storage, mycotoxins remain chemically stable, making strict environmental control essential to prevent further contamination [15]. The persistence of these toxins underscores the need for stringent regulatory compliance to ensure beer safety [15].

European regulations set maximum permissible levels for mycotoxins in food products, as established by Commission Recommendation 2023/915/EU [16]. Specifically, the limits for OTA are 3 µg/kg, aflatoxin B1 is restricted to 2 µg/kg, and the total sum of aflatoxins B1, B2, G1, and G2 must not exceed 4 µg/kg. These thresholds apply to cereal-based products, including those used in beer production. For DON, the maximum limit for unprocessed cereals (excluding durum wheat, oats, and corn) is 750 µg/kg, while for fumonisins (B1 and B2), the limit is 4000 µg/kg in unprocessed corn. Additionally, European Commission Regulation 2024/1038 establishes specific limits for T-2 and HT-2 toxins based on the type of raw material, with values ranging from 50 µg/kg to 1250 µg/kg [17]. Although these limits do not directly apply to beer, producers must closely monitor raw materials to ensure that mycotoxin levels remain within safe limits, thereby protecting consumer health.

Researchers continue to develop rapid and reliable methods for detecting mycotoxins in raw materials and final beer products [18]. High-performance liquid chromatography coupled with mass spectrometry (HPLC-MS) is a key technique for identifying and quantifying mycotoxins at trace levels. Although these methods require a higher capital investment compared to conventional HPLC-UV-RF instruments, their sensitivity and specificity are critical for ensuring regulatory compliance [19,20,21,22,23,24]. The use of an LC-QQQ mass analyzer enables precise ion mass measurements and allows for the identification of compounds potentially contributing to beer toxicity [25].

Innovative approaches, such as Vicam 6-in-1 immunoaffinity columns, enhance the detection and purification of multiple mycotoxins—including aflatoxins, OTA, DON, ZEN, and fumonisins—in complex brewing matrices. By streamlining analytical workflows, these columns reduce labor and minimize matrix interferences, ensuring compliance with safety regulations while protecting consumer health.

A new method using liquid chromatography/electrospray ionization tandem mass spectrometry (HPLC/ESI-MS/MS) has also been developed for the simultaneous qualitative and quantitative determination of nine mycotoxins. This technique incorporates a multianalyte immunoaffinity column (AOFZDT2^TM^, VICAM, Watertown, MA, USA), containing antibodies for aflatoxins (AFB1, AFB2, AFG1, AFG2), OTA, Fumonisins (FB1, FB2), DON, and HT-2, which allows for single-step extraction and cleanup (VICAM Watertown, MA, USA).

The application of Vicam 6-in-1 immunoaffinity columns, combined with HPLC-MS, has been extended to craft beers and freeze-dried beer samples to enhance analytical accuracy while minimizing false positives or negatives due to improper sample handling. Freeze-drying beer preserves sample integrity over time and reduces potential matrix effects that could interfere with mycotoxin detection [26]. By converting beer into a stable powdered form, degradation and contamination risks are significantly reduced, ensuring consistency in sample preparation and analysis [26]. This approach allows for accurate comparisons across various brewing styles and production methods.

The successful application of this method to freeze-dried beer confirms its robustness and adaptability, demonstrating that immunoaffinity columns effectively isolate multiple mycotoxins from complex matrices. Combined with the precision of HPLC-MS, this technique supports the reliable identification and quantification of mycotoxins in both liquid and freeze-dried beer samples. By mitigating sample variability and handling errors, this approach ensures accurate routine testing, improving quality control and safety in beer production.

These advancements demonstrate the ongoing efforts to enhance mycotoxin detection in beer, ensuring compliance with regulatory standards while improving consumer safety.

## 2. Materials and Methods

### 2.1. Chemical and Reagent

Acetonitrile, methanol, toluene (all HPLC grade), and glacial acetic acid were purchased from Mallinckrodt Baker (Milan, Italy). Ultrapure water was produced using a Milli-Q system (0.0056 µS/cm Millipore, Bedford, MA, USA). Ammonium acetate (for mass spectrometry), AFB1, AFB2, AFG1, AFG2, FB1, FB2, OTA, DON, and HT-2 were purchased from Sigma-Aldrich (Milan, Italy). Whatman^®^ glass microfiber grade GF/C filter discs (1.2 μm pore size, white, binder free, 0.26 mm thick, 100 ea, 47 mm diameter) were obtained from Whatman International (Maidstone, UK). DONtestTM HPLC, T-2testTM HPLC, and Myco6in1+ immunoaffinity columns (AOFZDT2^TM^, VICAM Watertown, MA, USA size 6ml) were obtained from VICAM (Watertown, MA, USA). Phosphate-buffered solution at pH 7.4 (PBS) was prepared by dissolving commercial phosphate-buffered saline tablets (Sigma-Aldrich) in ultrapure water.

#### Samples

The sample extraction and purification procedure were based on the work carried out by Lattanzio VM.T. et al., 2007 [27]. Three lager beers (pilsner style) produced in a pilot plant were used to develop the method. Beer sample stored at +4 °C were degassed (three cycles of 10 min each at +10 °C to avoid excessive sample heating). The same lager beer was also converted to a powder using freeze-drying processes (Alpha 1-2LDplus). The main drying time was set to 1 h under vacuum at −20 °C and 1.0 mbar.The final drying time was set until complete drying at vacuum −53 °C and 0.025 mbar.

### 2.2. Sample Preparation

The degassed beer samples (10 mL) were first extracted with 50 mL of PBS by shaking them in an orbital incubator Stuart SI500 at 150 rpm and 25 °C for 60 min. After centrifugation at 3000 rpm min^−1^ for 10 min at 25 °C, 35 mL of PBS extract (extract A) were collected and filtered through a glass microfiber filter (Whatman^®^ glass microfiber grade GF/C filter discs, 1.2 μm pore size, white, binder free, 0.26 mm thick, 100 ea, 47 mm diam). Then, 35 mL of methanol was added to the remaining solid material, containing 15 mL of PBS, and the sample was extracted again by shaking for 60 min. After centrifugation (3000 rpm min^−1^, 10 min), 10 mL of methanol/PBS extract were diluted with 90 mL of PBS to reduce the organic fraction and filtered through a glass microfiber filter (extract B). Aliquots of the two extracts were separately submitted to cleanup through the same multianalyte IAC. Then, 50 mL of extract B were passed through the myco 6-in-1 (AOFZDT2^TM^) column at 1–2 drops per second; the column was then washed with 20 mL of PBS to completely remove methanol residues. After passing 5 mL of extract A that was eluted at 1–2 drops per second, the column was washed with 10 mL of distilled water to remove PBS residue and matrix-interfering compounds. Toxins were eluted from the column with 3 mL of methanol in two steps of 1.5 mL of each at 1 drop per second. After the first step, a 5 min interval was allowed to favor methanol–antibody contact; complete elution of all toxins was obtained with the second elution step followed by air flushing through the column. The methanolic eluate was dried under an air stream at 50 °C and reconstituted with 250 µL of mobile phase. The sample was stored at −18° in the dark until analysis. Then, 100 µL were injected to be analyzed using LC/MS/MS. The same extraction procedure was carried out on the freeze-dried beer (500 mg of lyophilized sample was made up to volume with 5 mL of water). As a control, the same beer doped with known standard concentrations of mycotoxins (Ochratoxin A, Afla B1, Afla B2, Afla G1, Afla G2, Fumonisin B1, Fumonisin B2 and Deoxynivalenol) according with legal limit was also extracted. For HT-2, the spiking amount was chosen arbitrarily as 1000 µg/L because this mycotoxin level was not regulated.

### 2.3. Standard and Matrix-Matched Calibration

Stock solutions of each mycotoxin were prepared at different concentrations by diluting the solutions in the appropriate solvent. DON, HT-2 were dissolved in acetonitrile, AFs in toluene/acetonitrile 99:1 (*v/v*), OTA in toluene/acetic acid 99:1, and FBs in acetonitrile/water 1:1 (*v/v*). Subsequently, a mix containing all the mycotoxins to be analyzed at the maximum concentration allowed by legal limits was prepared. The mix was prepared with DON 1000 µg/L, HT-2 1000 µg/L, AFB1 5 µg/L, AFB2 5 µg/L, AFG1 5 µg/L, AFG2 5 µg/L, OTA 10 µg/L, FB1 500 µg/L, and FB2 500 µg/L diluted with mobile phase (A water and B methanol, both containing 1 mM of ammonium acetate and 0.1% acetic acid).

This mix was then diluted differently and used to prepare calibration solutions and the spiking solution. Calibration solutions for the standard calibration curves (at five points) were prepared in the LC mobile phase (A water and B methanol, both containing 1 mM of ammonium acetate and 0.1% acetic acid) by diluting appropriate amounts of the starting mix solution.

### 2.4. LC-ESI MS/MS Method

The Shimadzu Nexera X2 chromatograph coupled with LCMS-8050 Triple Quad LC/MS detector (Shimadzu Corporation, Milan, Italy) was used to separate mycotoxins. The UHPLC system consisted of a binary pump, automatic degasser, column heater, and autosampler. The MS/MS system was equipped with an ESI source operating in the positive ion mode according to the parameters shown in Table 1.

The chromatographic elution was conducted using a LUNA 3 µm C18 column (100 Å—size 150 × 3 mm) preceded by a Gemini C18 guard column (4 mm × 2 mm, 5 µm particles) as the stationary phase and a gradient H_2_O:MeOH (A water and B methanol, both containing 1 mM of ammonium acetate and 0.1% acetic acid) as the mobile phase (Table 2).

The optimization of ionization sources and MS parameters for each analyte was operated using the direct standard infusion of the standard solutions (Table 3).

### 2.5. LOD e LOQ

The method’s detectability was confirmed through the established limits of detection (LOD) and limits of quantification (LOQ) for each mycotoxin. LOD and LOQ were defined as the lowest concentrations of each mycotoxin that produced signal to noise ratios of 3:1 and 10:1, respectively.

### 2.6. Selectivity

The method’s selectivity was evaluated by analyzing blank samples and blank samples spiked with the target mycotoxins. This assessment was based on monitoring the characteristic multiple reaction monitoring (MRM) transitions for each mycotoxin at their specific retention times and verifying the relative response ratio between quantification and confirmation channels.

### 2.7. Optimization Method

The development of the method involved the extraction and purification of the mycotoxins and the optimization of the mass parameters followed by chromatographic optimization.

The MS detection was carried out in MRM mode, in order to obtain high sensitivity and selectivity for each analyte, based on the generation of protonated molecular ions from the source of the mycotoxins, as well as on the collision-induced production of fragments of specific ions of the mycotoxin. The specific MRM mycotoxins transition as well as their corresponding fragment voltages, collision energies, and dwell were determined individually for each analyte (Table 3). The precursor ions and the two most intense product ions for each analyte were measured for quantification and identification (confirmation), respectively.

The chromatographic method for simultaneous identification of mycotoxins has been refined by optimizing various analytical parameters, including the choice of columns, the mobile phase, flow rate, and oven temperature. Following this, parameters such as flow rate, oven temperature, and injection volume were fine tuned.

Chromatographic separation was further enhanced by adjusting the mobile phase composition, resulting in a gradient mode.

Our method achieved elution times of approximately 45 min for nine mycotoxins.

The assessment of potential interferences in compound quantification, arising from the direct injection of beer, was conducted through the evaluation of matrix effects (ME).

Five levels were utilized to create matrix calibration curves. ME was computed leveraging the slopes of the calibration curves in both solvent and matrix. The matrix effect remained well within the acceptable range of 80% to 120%.

Regarding the development of the multi-mycotoxin method in beer, the chromatographic results are shown in Figure 1a,b and Figure 2.

In detail, a method that allows, through a single analysis, the determination of nine mycotoxins—Ochratoxin A, Aflatoxins (B1, B2, G1, and G2), Fumonisins (B1 and B2), Deoxynivalenol, and HT-2—has been developed.

Five concentration levels of mycotoxins standards were prepared from a stock solution at different concentration ranges (Table 4). Five analyses were performed for each concentration level with the LC-MS 8050 system under optimized chromatographic conditions. By obtaining the equations for the regression line, five calibration curves were constructed using the least squares method. Mandel’s test proved that all calibration curves within the considered range were linear. The limits of quantification (LoQs) and limits of detection (LoDs) (Table 4) were calculated by multiplying the standard deviation (SD) of the lowest level of the calibration curve (n = 7) by ten and three, respectively, and dividing the result by the slope of the calibration curve.

The repeatability and reproducibility values (Table 4) were expressed as the percentage coefficient of variation (CV%) and calculated by dividing the corresponding standard deviations by average of the areas of the lowest level of the calibration curve (n = 5). Finally, the fourth level (n = 4) of each calibration curve (Table 5) was used to determine the retention time (RT), instrumental recovery, and relative standard deviation percentage (RSD%).

## 3. Results and Discussions

The study successfully developed and validated an innovative analytical method for the simultaneous detection of nine mycotoxins in craft beer, utilizing freeze-dried samples, an immunoaffinity column (IAC), and HPLC-ESI-MS/MS. The results demonstrated that the method is highly sensitive, specific, and reproducible, with limits of detection (LOD) and quantification (LOQ) below the maximum allowable levels established for raw materials by European regulations.

The use of freeze-dried beer significantly improved analytical performance by concentrating mycotoxins and reducing matrix interferences. This method enhances stability, allowing for more accurate quantification and comparison across different beer samples. The results showed that freeze-dried samples had lower variability compared to liquid samples, indicating that this approach can improve consistency in mycotoxin analysis.

The method allowed for the simultaneous detection of multiple mycotoxins in a single analysis, confirming their co-occurrence in beer. This is crucial, as mycotoxins can have additive or synergistic toxic effects, making multi-mycotoxin analysis essential for comprehensive risk assessment. The study found that certain mycotoxins, particularly deoxynivalenol (DON) and Fumonisins (FB1, FB2), were consistently detected, aligning with previous research indicating their prevalence in beer.

While European regulations specify limits for mycotoxins in cereals and raw materials, no direct legal thresholds exist for beer. However, the detected levels of mycotoxins in the tested craft beer samples remained within the safety margins when considering raw material limits. This finding underscores the importance of continuous monitoring and highlights the need for potential regulatory updates to include specific beer-related mycotoxin limits.

To ensure the reliability of the proposed method, validation was conducted in accordance with international guidelines. Key parameters such as linearity, sensitivity, accuracy, precision, and recovery were assessed. The method demonstrated excellent linearity across all mycotoxins, with R^2^ values exceeding 0.999, indicating a strong correlation between concentration and instrument response (Table 5). The LOD and LOQ values obtained were significantly lower than those of conventional HPLC-UV and LC-MS methods, highlighting the enhanced sensitivity of our approach.

Matrix effects, a critical consideration in complex beer samples, were evaluated by comparing standard calibration curves prepared in pure solvent versus beer matrix. Results indicated that the matrix effect remained within an acceptable range (80–120%), ensuring accurate quantification of mycotoxins. The method’s precision, assessed through intra- and inter-day repeatability, yielded relative standard deviations (RSD%) below 6% for all analytes, confirming its robustness and reproducibility.

The developed method was applied to craft beer samples to evaluate mycotoxin contamination levels. Table 6 presents the results obtained from both liquid and freeze-dried beer samples. The analyzed mycotoxins include HT-2, Fumonisins B1 (FB1), Fumonisins B2 (FB2), Deoxynivalenol (DON), Aflatoxins (AFB1, AFB2, AFG1, and AFG2) and Ochratoxin A (OTA). The freeze-dried beer samples tended to show slightly higher concentrations of mycotoxins compared to the liquid beer samples, likely due to the removal of water content which concentrates the analytes.

In particular, Deoxynivalenol (DON) was detected, with concentrations ranging from 10.25 to 11.14 µg/L in liquid beer samples and from 11.85 to 12.45 µg/L in freeze-dried beer samples. Fumonisins B1 and B2 were detected in varying concentrations, with FB1 ranging from 20.25 to 21.14 µg/L in liquid beer samples and from 22.85 to 24.65 µg/L in freeze-dried beer samples, and FB2 ranging from 52.78 to 59.98 µg/L in liquid beer samples and from 63.04 to 65.86 µg/L in freeze-dried beer samples. HT-2 showed the highest concentrations ranging from 86.85 to 90.18 µg/L in liquid beer samples and from 115.88 to 119.12 µg/L in freeze-dried beer samples.

It is important to note that Ochratoxin A (OTA) and Aflatoxins (AFTs) were not detected above the limit of quantification (LOQ) in any of the samples analyzed.

According to our study, DON was detected in Polish beers (12.3 μg kg^−1^) [28], in beers analyzed in Germany (7.1 μg kg^−1^) [29], and concentrations of 6.9 μg/L and 8.4 μg/L were reported by Varga et al. (2013) for Austrian and German beer, respectively [30]. In Veracruz, 87.5% of contaminated beer samples contained DON, according to Wall-Martinez et al. (2019) [31]. In contrast, in their study, Piacentini et al. (2015) [14] found deoxynivalenol (DON) to be the most common mycotoxin, with concentrations ranging from 127 to 501 µg/L.

Latvian beers have been found to contain the mycotoxins DON (73.2 μg/kg), HT-2 (2.0 μg/kg), FB1 (22.6 μg/kg), and AFB1 (0.14 μg/kg) [32].

According to our study, Piacentini et al. (2015) [14] reported Fumonisin FB1 concentrations ranging from 29 to 285 μg/L. The absence of aflatoxins and ochratoxin A (OTA) has been documented in several studies [31,33,34].

These results are in contrast with prior research indicating that wheat-based beers tend to exhibit higher DON contamination compared to barley-based varieties due to the greater susceptibility of wheat to Fusarium infections (Pascari et al., 2018) [15]. Moreover, the freeze-dried beer samples showed slightly higher mycotoxin concentrations than their liquid counterparts, likely due to the removal of water content, which effectively concentrated the analytes.

A comparison of the developed method with conventional approaches is presented in Table 7.

The developed method demonstrated superior sensitivity and throughput while maintaining cost-effectiveness. Unlike traditional HPLC-UV, which suffers from lower specificity, or standard LC-MS/MS, which requires extensive sample preparation, our approach streamlines the extraction and purification process using immunoaffinity columns (IAC) and freeze-drying, enhancing both efficiency and accuracy.

Additionally, the new approach offers broader applicability by enabling the simultaneous detection of nine mycotoxins in a single analysis. This advantage is particularly relevant for routine screening in the brewing industry, where multi-mycotoxin contamination is a concern.

The findings of this study emphasize the importance of continuous monitoring of mycotoxins in beer production. Given the persistence of these contaminants throughout the brewing process, breweries should implement stringent quality control measures, including raw material screening and optimized brewing practices, to minimize mycotoxin presence. Using freeze-dried beer instead of traditional liquid beer in mycotoxin analysis significantly enhances the efficiency of the extraction and purification process when utilizing VICAM columns. The freeze-drying process concentrates the analytes, reducing the sample’s water content and simplifying matrix handling. This results in improved recovery rates and cleaner extracts, allowing the VICAM columns to operate more effectively.

Freeze-drying offers several advantages over analyzing liquid beer, particularly when dealing with complex matrices like beer that can pose challenges in terms of liquid handling and matrix interference. One of the main reasons freeze-drying is superior is that it helps concentrate the sample, removing much of the water content and minimizing matrix effects that can interfere with the detection of mycotoxins. Beer contains various compounds such as sugars, proteins, and alcohol, which can affect the accuracy and sensitivity of analytical techniques like HPLC-ESI-MS/MS. These components can cause ion suppression or enhance matrix effects, making it difficult to reliably detect mycotoxins in a liquid sample.

Furthermore, freeze-drying also enhances the stability of the sample, preventing degradation of mycotoxins over time, which could be a concern when dealing with liquid beer samples.

The LC-MS/MS method developed using IAC and freeze-dried beer samples represents a significant advancement in mycotoxin analysis. By offering enhanced sensitivity, reduced matrix effects, and improved efficiency, this approach provides a robust tool for beer quality control. Future studies should focus on expanding the dataset by analyzing a broader range of beer styles and investigating the impact of different brewing conditions on mycotoxin stability.

## 4. Conclusions

Mycotoxin contamination poses a significant challenge to food safety, especially in the craft beer industry, which is growing in popularity. To address this issue, a new analytical method for the simultaneous detection of nine mycotoxins in beer was developed using liquid chromatography coupled with tandem mass spectrometry (LC-MS/MS). This method offers a rapid, sensitive, and efficient approach to mycotoxin analysis, significantly enhancing the ability to monitor co-occurring mycotoxins throughout the beer production process. The incorporation of VICAM 6-in-1 columns improves precision and efficiency, allowing breweries to conduct comprehensive screenings of raw materials such as barley, hops, and malt, as well as monitor contamination at critical production stages like malting and storage. By ensuring that mycotoxin levels remain within safe limits, this method safeguards consumer health and helps breweries comply with regulatory standards. Although specific legal thresholds for mycotoxins in beer are not universally established, the method ensures compliance with limits set for raw materials and cereals, helping breweries avoid legal risks and market recalls. Economically, the method reduces costs by streamlining sample preparation, using freeze-drying to minimize matrix interferences, and enabling simultaneous detection of multiple mycotoxins. This decreases reagent usage, shortens analysis time, and eliminates the need for separate toxin-specific tests. Large-scale breweries and regulatory bodies can use this method to monitor supply chains and ensure product safety for domestic and international markets.

The study highlights the importance of continuous monitoring and advocates for updating regulations to include specific mycotoxin limits for beer. Future research is recommended to expand the method’s application to other beer styles and ingredients and investigate the effects of brewing processes on mycotoxin stability. This approach represents a significant step forward in ensuring food safety, maintaining consumer trust, and supporting the growth of the craft beer industry.

## Figures and Tables

**Figure 1 foods-14-00956-f001:**
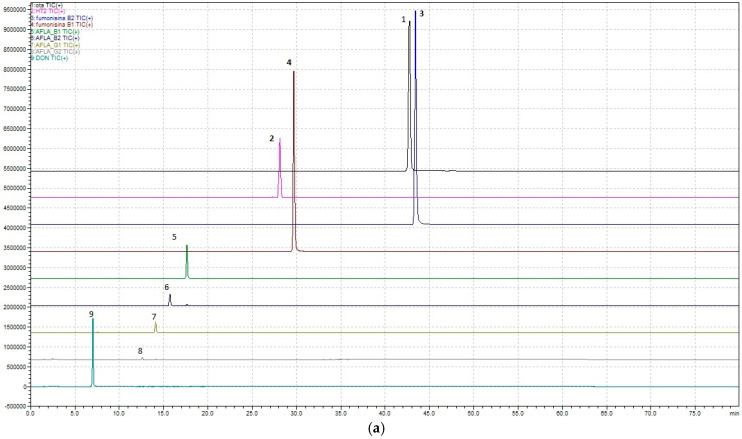
(**a**) SCAN profile for freeze-dried doped beer (obtained by adding the standard blend to the freeze-dried model beer). 1. Ochratoxin A; 2. HT2; 3. Fumonisin B2; 4. Fumonisin B1; 5. Aflatoxin B1; 6. Aflatoxin B2; 7. Aflatoxin G1; 8. Aflatoxin G2; 9. Deoxynivalenol. (**b**) SCAN doped beer with standards mix. 1. Ochratoxin A; 2. HT2; 3. Fumonisin B2; 4. Fumonisin B1; 5. Aflatoxin B1; 6. Aflatoxin B2; 7. Aflatoxin G1; 8. Aflatoxin G2; 9. Deoxynivalenol.

**Figure 2 foods-14-00956-f002:**
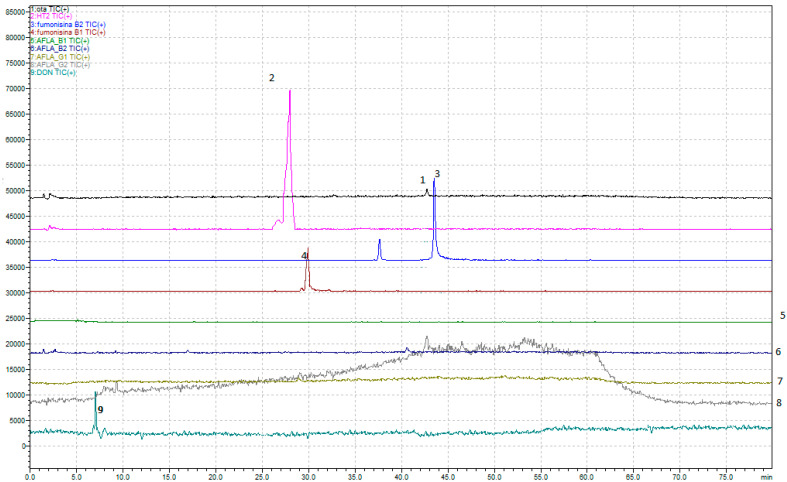
SCAN profile for undoped beer. 1. Ochratoxin A; 2. HT2; 3. Fumonisin B2; 4. Fumonisin B1; 5. Aflatoxin B1; 6. Aflatoxin B2; 7. Aflatoxin G1; 8. Aflatoxin G2; 9. Deoxynivalenol.

**Table 1 foods-14-00956-t001:** LC-ESI-MS/MS condition.

Compounds	RT
Nebulizing Gas Flow L/min	3
Heating Gas Flow L/min	10
Interface Temperature °C	300
DL * Temperature °C	250
Het Bloc Temperature °C	400
Drying Gas Flow L/min	10

* DL: desolvatation line.

**Table 2 foods-14-00956-t002:** Concentration linear gradient.

	Gradient	
Time	%B *	Value
0.20	B conc	20
3.00	B Conc.	40
40.00	B Conc.	63
58.00	B Conc.	63
60.00	Stop	-

* %A = 100-%B.

**Table 3 foods-14-00956-t003:** Fragmentation mycotoxins.

Compounds	Precursor (m/z)	Product (m/z)	Dwell (m sec)	Q1 Prebias (Volt)	CE	Q3 Prebias (Volt)
OTA	403.8	239.15	100	−14	−24	−18
	403.8	357.95	100	−14	−15	−28
HT-2	424	263.2	100	−12	−15	−14
	424	157.05	100	−12	−23	−12
Fumonisin B2	706.1	336.3	100	−20	−38	−18
	706.1	318.4	100	−20	−41	−17
Fumonisin B1	722.1	334.2	100	−20	−41	−18
	722.1	352.35	100	−20	−37	−19
Aflatoxin B1	312.8	241	100	−15	−38	−18
	312.8	185.2	100	−14	−51	−21
Aflatoxin B2	314.8	287	100	−11	−25	−22
	314.8	259.1	100	−11	−30	−20
Aflatoxiin G1	328.8	311	100	−11	−21	−24
	328.8	245.1	100	−11	−28	−18
Aflatoxin G2	331	313.1	100	−11	−11	−17
	330.8	245.1	100	−15	−30	−19
DON	296.9	249.05	100	−13	−11	−19
	269.9	231	100	−13	−11	−22

**Table 4 foods-14-00956-t004:** Chromatographic optimization.

Compounds	Linear Range (ug/L)	R^2^	LOD (ug/L)	LOQ (ug/L)	Repeatability (%)	Riproducibility (%)
Ochratoxin A	2–10	0.9998	0.423	1.233	1.91	1.49
Aflatoxin B1	1–5	0.9996	0.362	0.863	3.56	3.42
Aflatoxin B2	1–5	0.9998	0.321	0.842	4.892	2.89
Aflatoxin G1	1–5	0.9997	0.415	0.950	5.68	4.30
Aflatoxin G2	1–5	0.9999	0.521	1.005	4.72	3.50
Fumonisin B1	10–500	0.9996	1.110	3.180	5.16	5.19
Fumonisin B2	10–500	0.9997	0.426	2.171	5.41	3.82
Deoxynivalenol	10–1000	0.9994	4.298	9.081	1.71	6.00
HT-2	10–1000	0.9996	1.910	6.264	5.57	4.19

**Table 5 foods-14-00956-t005:** Retention time. instrumental recovery, and relative standard deviation percentage of mycotoxins.

Compounds	RT	Concentration (ug/L)	RSD%	Recovery (%)
Ochratoxin A	43.08	8	11	82
Aflatoxin B1	17.55	4	10	104
Aflatoxin B2	15.48	4	6	98
Aflatoxin G1	14.01	4	6	102.7
Aflatoxin G2	12.42	4	7	95
Fumonisin B1	28.89	400	13	101
Fumonisin B2	42.96	400	8	96
Deoxynivalenol	44.05	800	0.2	79
HT-2	28.15	800	1	180

**Table 6 foods-14-00956-t006:** Quantitative mycotoxin contamination levels to craft beers and freeze-dried beer.

	Beer (µg/L)			Freeze-Dried Beer (µg/L)		
	1	2	3	1	2	3
HT2	86.85 ± 4.92	90.18 ± 14.56	88.11 ± 7.38	115.88 ± 5.45	107.03 ± 13.21	119.12 ± 11.95
FUMB2	55.79 ± 3.03	52.78 ± 3.21	59.98 ± 4.22	65.86 ± 2.13	64.57 ± 4.21	63.04 ± 5.76
FUMB1	21.14 ± 0.93	20.25 ± 0.89	20.79 ± 0.49	23.45 ±1.03	24.65 ± 1.17	22.85 ± 0.36
DON	11.14 ± 0.93	10.25 ± 0.89	10.79 ± 0.49	12.45 ±1.03	12.65 ± 1.17	11.85 ± 0.36
OTA	<LOQ	<LOQ	<LOQ	<LOQ	<LOQ	<LOQ
AFTs	<LOQ	<LOQ	<LOQ	<LOQ	<LOQ	<LOQ

**Table 7 foods-14-00956-t007:** Comparison of the different published methods for multi-mycotoxin determination.

Method	Sample Preparation	Matrix	Sensitivity	Specificity	Cost-Effectiveness	Multi-Mycotoxin Detection	Citation
**HPLC/ESI-MS/MS with IAC**	IAC + Freeze-Drying	Beer and freeze-dried beer	High	High	Cost-Effective	Yes (9 Mycotoxins)	This study
**LC/QTOF-MS/MS**	QuEChERS	Malted barley and beer	High	High	Moderate	Yes	Lago et al. (2020) [35].
**HPLC-TOF-MS with QuEChERS**	QuEChERS—SPE extraction	Beer	High	High	Moderate	Yes (11)	Bogdanova et al. (2018) [32].
**UPLC-TOF-HRMS**	IAC	Malted barley and beer	High	High	High	Yes (DON and NIV)	Edyta Ksieniewicz-Woźniak et al. (2019) [28].
**LC-MS/MS**	Reconstitution of the dried-down sample in solvent	Beer	High	High	High	Yes (deoxynivalenol-3-glucoside and 3-acetyl-deoxynivalenol)	Varga et al. (2013) [30].
**UHPLC-MS/MS applying the**	SPE column	Beer	High	High	Moderate	Yes (23 Mycotoxins)	Hiram A Wall-Martínez (2019) [31].
**Monte Carlo method**							
**GC-MS/MS**	QuEChERS—Dispersive solidphase extraction (d-SPE)	Beer	High	High	Moderate	Yes (7 Mycotoxins)	Rodríguez-Carrasco Y et al. (2015) [33].
**LC-HRMS**	QuEChERS—SPE	Beer	High	High	High	Yes (7 Mycotoxins)	Irina Rozentale (2018) [34].
**LC-MS/MS**	SPE	beer	High	High	Moderate	Yes (4 Mycotoxins)	Katharina Habler (2017) [29].
**LC-MS/MS**	IAC	Cereals, tea and herbal infusions	High	High	Moderate	Yes	Gonçalves et al. (2020) [36].
**LC-QTOF-MS with SPE Cleanup**	Requires specialized SPE sorbents	Cereals and cereal-based food	Very High	High	High	Yes (Broad Range of Mycotoxins)	Malachová et al. (2018) [19].
**HPLC-UV**	Extensive sample preparation needed	Maize	Moderate	Low	Moderate	Limited	Leite et al. (2020) [37].

## Data Availability

Data are contained within the article.

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
