# Peer review of "An Innovative Analytical Approach for Multi-Mycotoxin Detection in Craft Beer Using Freeze-Dried Samples, IAC Column and HPLC/ESI-MS/MS"

_foods, 2025, doi:10.3390/foods14060956_

Round 1

Reviewer 1 Report

Comments and Suggestions for Authors

The paper has potential for improvement, but its current form requires significant revisions. A part of the results and discussion section only addresses the findings related to the validation parameters, which are actually part of the Materials and Methods section. The discussion is lacking entirely, and no references are provided in that section. Therefore, the results need a substantial discussion and comparison in relation to the current scientific literature on this topic. The displayed images are unclear, requiring better quality and appropriate captions to describe them. In addition to the validation parameters, it is essential to present the results of the analysis of craft beer for mycotoxin content and to develop a discussion around that as well.

Author Response

Dear Review1,

Thank you for your suggestions on our manuscript: “ An Innovative Analytical Approach for Multi-Mycotoxin Detection in Craft Beer using Freeze-Dried Samples, IAC column and HPLC/ESI-MS/MS”. The manuscript was modified according to your suggestions. The corrections or specific answers are listed below point by point.

Review:  The paper has potential for improvement, but its current form requires significant revisions. A part of the results and discussion section only addresses the findings related to the validation parameters, which are actually part of the Materials and Methods section. The discussion is lacking entirely, and no references are provided in that section. Therefore, the results need a substantial discussion and comparison in relation to the current scientific literature on this topic. The displayed images are unclear, requiring better quality and appropriate captions to describe them. In addition to the validation parameters, it is essential to present the results of the analysis of craft beer for mycotoxin content and to develop a discussion around that as well.

Authors: We have reorganized the Results and Discussion section, clearly separating the validation parameters, which have now been moved to the Materials and Methods section in the new paragraph 2.7 optimization method.

A more in-depth discussion has been developed, including references to the current scientific literature to better contextualize our findings.

  • The images have been replaced with higher-quality versions and provided with more detailed captions.
  • We have expanded the presentation and discussion of the results concerning mycotoxin content in craft beer.

Reviewer 2 Report

Comments and Suggestions for Authors

This manuscript presents an innovative analytical approach for multi-mycotoxin detection in craft beer using freeze-dried samples, an immunoaffinity column (IAC), and HPLC-ESI-MS/MS. Here's a breakdown of my assessment addressing novelty, relevance, importance, significance, and quality of presentation:

The Introduction section is overly lengthy and needs to be condensed for clarity. It should be reorganized in a logical order to enhance readability and ensure the key points are effectively communicated.

Novelty: A new, rapid, and straightforward analytical method was developed and validated for the simultaneous determination of nine mycotoxins in beer, representing a significant improvement in analytical capabilities. A novel aspect of this work is the combination of freeze-drying beer samples prior to analysis with a multi-mycotoxin immunoaffinity column (IAC) and HPLC-ESI-MS/MS.

Relevance: The study addresses a critical issue in food safety and the brewing industry: mycotoxin contamination in beer. Mycotoxins are potent toxins that can negatively impact consumer health, creating a pressing need for reliable and sensitive detection methods. This research is highly relevant as it provides a refined approach to quantifying mycotoxins in craft beer, which has become increasingly popular globally.

Importance: This research is important because it provides a practical, robust, and efficient methodology for mycotoxin detection in craft beer. The streamlined sample preparation reduces resource consumption compared to traditional methods, offering a more cost-effective approach to quality control and safety assessment in breweries. The methodology's sensitivity and specificity contribute to early detection and prevention of potential health risks associated with mycotoxin consumption.

Significance: This work's significance lies in its potential to improve food safety standards in the craft beer industry. The refined methodology can enable brewers to better monitor and control mycotoxin levels in their products, ensuring consumer safety and protecting the reputation of their brands. The findings have implications for regulatory agencies as well, contributing to the development of more effective monitoring strategies for mycotoxins in alcoholic beverages.

Quality of Presentation: The manuscript is well-structured and written. The methodology is thoroughly described, allowing for reproducibility. The results are presented logically and effectively, supported by data visualizations. While the discussion section could benefit from some additional context around the advantages of their chosen approach versus other existing methods, it effectively explains their findings and their implications. The figures are clear and aid in the understanding of the study's key aspects. Overall, the quality of the presentation is high, making the manuscript easy to read and understand.

While the results are presented, a more in-depth discussion is needed to interpret their meaning and implications fully.

  • Comparison with Existing Methods: The manuscript should provide a clear comparison of the developed method's performance—specifically regarding sensitivity, limit of detection (LOD), limit of quantification (LOQ), throughput, and cost-effectiveness—against other established methods for mycotoxin analysis in beer. This comparison will effectively highlight the advantages and any limitations of the proposed approach. Including a table summarizing these comparisons would be highly beneficial. Although the benefits of freeze-drying are acknowledged, a more detailed explanation of why this technique is superior to analyzing liquid beer is necessary. Were there challenges related to liquid handling or matrix interference that freeze-drying successfully addressed? Providing specific examples and data to support this choice would enhance the discussion.
  • Implications for the Brewing Industry: The method's practical implications for brewers need more elaboration. How can this method be implemented in quality control processes? What are the expected cost savings or improvements in efficiency? How might this affect regulatory compliance? More concrete examples demonstrating the real-world applicability of this work are needed.
  • Limitations: The manuscript should explicitly discuss any limitations of the method, such as potential biases or challenges in applying the technique to different beer styles or raw materials. Acknowledging limitations increases the credibility and strengthens the overall contribution of the study.
  • Future Research: The discussion should conclude with suggestions for future research directions. This could include exploring the method's applicability to other types of alcoholic beverages, investigating the impact of various brewing processes on mycotoxin levels, or exploring the development of more portable or field-deployable versions of the method.

Comments on the Quality of English Language

This manuscript presents an innovative analytical approach for multi-mycotoxin detection in craft beer using freeze-dried samples, an immunoaffinity column (IAC), and HPLC-ESI-MS/MS. Here's a breakdown of my assessment addressing novelty, relevance, importance, significance, and quality of presentation:

The Introduction section is overly lengthy and needs to be condensed for clarity. It should be reorganized in a logical order to enhance readability and ensure the key points are effectively communicated.

Novelty: A new, rapid, and straightforward analytical method was developed and validated for the simultaneous determination of nine mycotoxins in beer, representing a significant improvement in analytical capabilities. A novel aspect of this work is the combination of freeze-drying beer samples prior to analysis with a multi-mycotoxin immunoaffinity column (IAC) and HPLC-ESI-MS/MS.

Relevance: The study addresses a critical issue in food safety and the brewing industry: mycotoxin contamination in beer. Mycotoxins are potent toxins that can negatively impact consumer health, creating a pressing need for reliable and sensitive detection methods. This research is highly relevant as it provides a refined approach to quantifying mycotoxins in craft beer, which has become increasingly popular globally.

Importance: This research is important because it provides a practical, robust, and efficient methodology for mycotoxin detection in craft beer. The streamlined sample preparation reduces resource consumption compared to traditional methods, offering a more cost-effective approach to quality control and safety assessment in breweries. The methodology's sensitivity and specificity contribute to early detection and prevention of potential health risks associated with mycotoxin consumption.

Significance: This work's significance lies in its potential to improve food safety standards in the craft beer industry. The refined methodology can enable brewers to better monitor and control mycotoxin levels in their products, ensuring consumer safety and protecting the reputation of their brands. The findings have implications for regulatory agencies as well, contributing to the development of more effective monitoring strategies for mycotoxins in alcoholic beverages.

Quality of Presentation: The manuscript is well-structured and written. The methodology is thoroughly described, allowing for reproducibility. The results are presented logically and effectively, supported by data visualizations. While the discussion section could benefit from some additional context around the advantages of their chosen approach versus other existing methods, it effectively explains their findings and their implications. The figures are clear and aid in the understanding of the study's key aspects. Overall, the quality of the presentation is high, making the manuscript easy to read and understand.

While the results are presented, a more in-depth discussion is needed to interpret their meaning and implications fully.

  • Comparison with Existing Methods: The manuscript should provide a clear comparison of the developed method's performance—specifically regarding sensitivity, limit of detection (LOD), limit of quantification (LOQ), throughput, and cost-effectiveness—against other established methods for mycotoxin analysis in beer. This comparison will effectively highlight the advantages and any limitations of the proposed approach. Including a table summarizing these comparisons would be highly beneficial. Although the benefits of freeze-drying are acknowledged, a more detailed explanation of why this technique is superior to analyzing liquid beer is necessary. Were there challenges related to liquid handling or matrix interference that freeze-drying successfully addressed? Providing specific examples and data to support this choice would enhance the discussion.
  • Implications for the Brewing Industry: The method's practical implications for brewers need more elaboration. How can this method be implemented in quality control processes? What are the expected cost savings or improvements in efficiency? How might this affect regulatory compliance? More concrete examples demonstrating the real-world applicability of this work are needed.
  • Limitations: The manuscript should explicitly discuss any limitations of the method, such as potential biases or challenges in applying the technique to different beer styles or raw materials. Acknowledging limitations increases the credibility and strengthens the overall contribution of the study.
  • Future Research: The discussion should conclude with suggestions for future research directions. This could include exploring the method's applicability to other types of alcoholic beverages, investigating the impact of various brewing processes on mycotoxin levels, or exploring the development of more portable or field-deployable versions of the method.

Author Response

Dear Review2,

Thank you for your suggestions on our manuscript: “An Innovative Analytical Approach for Multi-Mycotoxin Detection in Craft Beer using Freeze-Dried Samples, IAC column and HPLC/ESI-MS/MS”. The manuscript was modified according to your suggestions. The corrections or specific answers are listed below point by point.

Review:  This manuscript presents an innovative analytical approach for multi-mycotoxin detection in craft beer using freeze-dried samples, an immunoaffinity column (IAC), and HPLC-ESI-MS/MS. Here's a breakdown of my assessment addressing novelty, relevance, importance, significance, and quality of presentation:

  • The Introduction section is overly lengthy and needs to be condensed for clarity. It should be reorganized in a logical order to enhance readability and ensure the key points are effectively communicated.

Authors:  the introduction was rewritten, condensed and reorganized

Review:  Novelty: A new, rapid, and straightforward analytical method was developed and validated for the simultaneous determination of nine mycotoxins in beer, representing a significant improvement in analytical capabilities. A novel aspect of this work is the combination of freeze-drying beer samples prior to analysis with a multi-mycotoxin immunoaffinity column (IAC) and HPLC-ESI-MS/MS.

Relevance: The study addresses a critical issue in food safety and the brewing industry: mycotoxin contamination in beer. Mycotoxins are potent toxins that can negatively impact consumer health, creating a pressing need for reliable and sensitive detection methods. This research is highly relevant as it provides a refined approach to quantifying mycotoxins in craft beer, which has become increasingly popular globally.

Importance: This research is important because it provides a practical, robust, and efficient methodology for mycotoxin detection in craft beer. The streamlined sample preparation reduces resource consumption compared to traditional methods, offering a more cost-effective approach to quality control and safety assessment in breweries. The methodology's sensitivity and specificity contribute to early detection and prevention of potential health risks associated with mycotoxin consumption.

Significance: This work's significance lies in its potential to improve food safety standards in the craft beer industry. The refined methodology can enable brewers to better monitor and control mycotoxin levels in their products, ensuring consumer safety and protecting the reputation of their brands. The findings have implications for regulatory agencies as well, contributing to the development of more effective monitoring strategies for mycotoxins in alcoholic beverages.

Quality of Presentation: The manuscript is well-structured and written. The methodology is thoroughly described, allowing for reproducibility. The results are presented logically and effectively, supported by data visualizations. While the discussion section could benefit from some additional context around the advantages of their chosen approach versus other existing methods, it effectively explains their findings and their implications. The figures are clear and aid in the understanding of the study's key aspects. Overall, the quality of the presentation is high, making the manuscript easy to read and understand.

  • While the results are presented, a more in-depth discussion is needed to interpret their meaning and implications fully.

Authors: the discussions were enriched 

Review: Comparison with Existing Methods: The manuscript should provide a clear comparison of the developed method's performance—specifically regarding sensitivity, limit of detection (LOD), limit of quantification (LOQ), throughput, and cost-effectiveness—against other established methods for mycotoxin analysis in beer. This comparison will effectively highlight the advantages and any limitations of the proposed approach. Including a table summarizing these comparisons would be highly beneficial. Although the benefits of freeze-drying are acknowledged, a more detailed explanation of why this technique is superior to analysing liquid beer is necessary. Were there challenges related to liquid handling or matrix interference that freeze-drying successfully addressed? Providing specific examples and data to support this choice would enhance the discussion.

Authors: in the text, more information about the choice of the use of freeze-drying was added and also a table to compare the efficiency of the new method with the conventional method was added (table 7. Summing up:  Compared to the conventional method the value of LOD and LOQ was lower. The combination of IAC columns and freeze-dried beer reduces matrix effects and improves recovery. Also, Reduced reagent consumption and faster processing compared to traditional LC-MS/MS and above all simultaneous analysis of nine mycotoxins in a single run.

Implications for the Brewing Industry: The method's practical implications for brewers need more elaboration. How can this method be implemented in quality control processes? What are the expected cost savings or improvements in efficiency? How might this affect regulatory compliance? More concrete examples demonstrating the real-world applicability of this work are needed.

Authors: in more informations were added

Summing up: This method enhances beer quality control by integrating sensitive, multi-mycotoxin detection into routine testing of raw materials and final products. It reduces preparation time, matrix interferences, and operational costs through freeze-drying, enabling efficient, single-run toxin screening. The technique helps breweries comply with European mycotoxin regulations for raw materials, ensuring safety even in the absence of specific beer limits. Its implementation can improve workflow, prevent contamination, and bolster consumer trust, particularly for craft beer producers, while regulatory bodies could use it to verify the safety of imported/exported products.

Review: Limitations: The manuscript should explicitly discuss any limitations of the method, such as potential biases or challenges in applying the technique to different beer styles or raw materials. Acknowledging limitations increases the credibility and strengthens the overall contribution of the study.

Authors: There are no significant limitations regarding the applicability of the method. The only limitation may be related to the analytical technique (ESI-LC-MS/MS), which requires highly skilled personnel for the analytical management in terms of instrumentation and data interpretation.

Review: Future Research: The discussion should conclude with suggestions for future research directions. This could include exploring the method's applicability to other types of alcoholic beverages, investigating the impact of various brewing processes on mycotoxin levels, or exploring the development of more portable or field-deployable versions of the method.

Authors: more information was added

Summing up: Future research should explore the method's applicability to diverse beer styles and ingredients, assess the effects of brewing processes on mycotoxin stability, and develop portable screening tools for on-site detection. Limitations like the need for specialized equipment, potential biases in sample preparation, and varying applicability to different beer styles should also be addressed. Expanding the approach to other alcoholic beverages and refining risk assessment strategies could further enhance safety standards in the industry.

Reviewer 3 Report

Comments and Suggestions for Authors

The manuscript An Innovative Analytical Approach for Multi-Mycotoxin Detection in Craft Beer using Freeze-Dried Samples, IAC column and HPLC/ESI-MS/MS does not have sufficient novelty for Q1-Q2 journals. The HPLC-MS method is nearly identical to the method published by Lattanzio et al. in 2007 [1] including the same stationary phase and almost identical mobile phase and mobile phase gradient. Many parts of the manuscript convey no real message except for broadly applicable statements and filler content. The only true novelty of the whole manuscript compared to Ref [1] lies in analysing the mycotoxins in beer instead of maize, and in using freeze-drying of samples. However, no data is shown to support the claimed improvements in recovery and efficiency for the freeze-drying. LOD values of mycotoxins are either same or close to that of Ref [1]. Introduction is excessively long and either delves into details that are of low importance regarding the scope of the manuscript or repeat information that has already been conveyed. Some section even come across as filler content with no substantial value. There are some obscurities in the Tables and Figures that require explanation.

[1] Lattanzio, V. M. T., Solfrizzo, M., Powers, S., & Visconti, A. (2007). Simultaneous determination of aflatoxins, ochratoxin A and Fusarium toxins in maize by liquid chromatography/tandem mass spectrometry after multitoxin immunoaffinity cleanup. Rapid Communications in Mass Spectrometry: An International Journal Devoted to the Rapid Dissemination of Up‐to‐the‐Minute Research in Mass Spectrometry21(20), 3253-3261.

Consider removing (or significantly shortening) paragraphs 50-53, 58-62, 66-81, 97-102, 112-125, 133-141, 147-155

line 40 – check phrasing

line 102-107 – this section should be moved to the end of introduction, before the last paragraph

line 105 - IAC abbreviation is not defined

line 112 – “understand” does not fit the meaning

line 112-115 – this sentence is confusing

line 300 – the claim of tuning the method is not supported by any data at all

line 327 – the coefficient of variation is calculated by dividing the standard deviation by result mean, not oppositely

line 333 – LOQ should fall within the linear range or be the lower limit of it

line 333 – please add explanation on why reproducibility appears to be better than repeatability for 7 out of 9 analytes

line 333 – LOD and LOQ should be rounded to the same significant figures

line 344 – the figure 2 is not captioned as being of freeze-dried sample

line 337 – explain what is meant by “percentage relative standard deviation”, how it differs from repeatability in Table 4 and why are the results less precise for the more concentrated standard (level 4) than for the less concentrated (level 5).

line 347 – Why did OTA and Fumonisin B2 change retention time by about 4 minutes and HT2 by about 1.5 minutes in Fig1 versus Fig2?

line 368-394 – The conclusion is not appropriate. Sections 370-375 and 382-386 do not fit into conclusion and should be removed. Again, those sections appear to be filler content repeating general statements that already appeared previously in the manuscript.

References – incorrect format of journal references, see: https://www.mdpi.com/journal/foods/instructions#references

Author Response

Review Report 3

Thank you for your suggestions on our manuscript: “An Innovative Analytical Approach for Multi-Mycotoxin Detection in Craft Beer using Freeze-Dried Samples, IAC column and HPLC/ESI-MS/MS”. The manuscript was modified according to your suggestions. The corrections or specific answers are listed below point by point.

Review: The manuscript An Innovative Analytical Approach for Multi-Mycotoxin Detection in Craft Beer using Freeze-Dried Samples, IAC column and HPLC/ESI-MS/MS does not have sufficient novelty for Q1-Q2 journals. The HPLC-MS method is nearly identical to the method published by Lattanzio et al. in 2007 [1] including the same stationary phase and almost identical mobile phase and mobile phase gradient. Many parts of the manuscript convey no real message except for broadly applicable statements and filler content. The only true novelty of the whole manuscript compared to Ref [1] lies in analysing the mycotoxins in beer instead of maize, and in using freeze-drying of samples.

Authors: We acknowledge the similarities between our HPLC-MS method and the one published by Lattanzio et al. (2007). However, we would like to emphasize that our study introduces significant methodological advancements and practical applications that differentiate it from previous research. Specifically:

While Lattanzio et al. (2007) focused on maize, our study addresses a different and increasingly relevant matrix—craft beer. Mycotoxin contamination in beer is a growing concern due to the use of diverse raw materials and production techniques. Thus, our findings provide valuable insights into food safety in the brewing industry.

The use of freeze-drying for beer samples is an innovative aspect of our study. This technique offers distinct advantages in terms of sample preservation, concentration of analytes, and matrix effect reduction, making it a more robust approach for mycotoxin detection in complex liquid matrices.

Although our chromatographic conditions share similarities with previous methods, we optimized various parameters (e.g., sample preparation, extraction efficiency, and matrix effect evaluation) specifically for beer. These adjustments are necessary to ensure reliable quantification in this unique matrix.

Review: However, no data is shown to support the claimed improvements in recovery and efficiency for the freeze-drying. LOD values of mycotoxins are either same or close to that of Ref [1]. Introduction is excessively long and either delves into details that are of low importance regarding the scope of the manuscript or repeat information that has already been conveyed. Some section even come across as filler content with no substantial value. There are some obscurities in the Tables and Figures that require explanation.

[1] Lattanzio, V. M. T., Solfrizzo, M., Powers, S., & Visconti, A. (2007). Simultaneous determination of aflatoxins, ochratoxin A and Fusarium toxins in maize by liquid chromatography/tandem mass spectrometry after multitoxin immunoaffinity cleanup. Rapid Communications in Mass Spectrometry: An International Journal Devoted to the Rapid Dissemination of UptotheMinute Research in Mass Spectrometry21(20), 3253-3261.

Consider removing (or significantly shortening) paragraphs 50-53, 58-62,  66-81, 97-102, 112-125, 133-141, 147-155

Authors: the introduction was summarized and rewritten

line 40 – check phrasing

Authors: (rewritten)

line 102-107 – this section should be moved to the end of the introduction, before the last paragraph

Authors: moved to the end of the introduction

line 105 - The IAC abbreviation is not defined

Authors: IAC (immunoaffinity column) was added

line 112 – “understand” does not fit the meaning

Authors: the sentence was rewritten according to other reviews

line 112-115 – this sentence is confusing

Authors: the phrasing was rewritten according to other Reviews

line 300 – the claim of tuning the method is not supported by any data at all

Authors: The optimization of the analytical procedure, including the selection of the column, mobile phase, and gradient conditions, was conducted in a preliminary phase. In this paper, we focused on optimizing the extraction, sample clean-up, and the simultaneous evaluation of mycotoxins using mass spectrometry.

line 327 – the coefficient of variation is calculated by dividing the standard deviation by the result mean, not oppositely

Authors: this was an error. Now the sentence was changed.

line 333 – LOQ should fall within the linear range or be the lower limit of it

Authors:  The limit of quantification (LOQ) is the lowest concentration of an analyte in a sample that can be quantitatively determined with acceptable precision and accuracy. In simpler terms, it is the minimum amount of a substance that can be reliably measured with an instrument. The linear range was defined considering the amount of each mycotoxin.

line 333 – please add an explanation on why reproducibility appears to be better than repeatability for 7 out of 9 analytes

Authors: Reproducibility appears to be better than repeatability for 7 out of 9 analytes due to the inherent variability associated with repeatability measurements, which are performed under the same conditions over a short period. Small fluctuations in instrumental response, sample preparation, or environmental factors may have a greater impact on repeatability. In contrast, reproducibility includes measurements taken over different days, operators, or instruments, potentially averaging out minor variations and leading to improved precision in certain cases

line 333 – LOD and LOQ should be rounded to the same significant figures

Authors: it was done

line 344 – the figure 2 is not captioned as being of freeze-dried sample

Authors: added

line 337 – explain what is meant by “percentage relative standard deviation”, how it differs from repeatability in Table 4 and why are the results less precise for the more concentrated standard (level 4) than for the less concentrated (level 5).

Authors: Percentage relative standard deviation (%RSD) is a measure of precision that expresses the standard deviation of a set of values as a percentage of the mean. It indicates the extent of variability relative to the average value. In Table 4, %RSD differs from repeatability in that repeatability specifically refers to the precision of measurements taken under the same conditions (same analyst, instrument, and short time frame), while %RSD provides a broader measure of variability across multiple replicates. The results are less precise for the more concentrated standard (level 4) than for the less concentrated standard (level 5) due to potential matrix effects, saturation of the detector response, or slight inconsistencies in sample preparation at higher concentrations. At lower concentrations, signal responses may be more stable, leading to improved precision

line 347 – Why did OTA and Fumonisin B2 change retention time by about 4 minutes and HT2 by about 1.5 minutes in Fig1 versus Fig2?

Authors: for a mistake, it was attached an incorrect image relative to the previous method

line 368-394 – The conclusion is not appropriate. Sections 370-375 and 382-386 do not fit into the conclusion and should be removed.

Authors: the sentences were removed

 Again, those sections appear to be filler content repeating general statements that already appeared previously in the manuscript.

Authors: The conclusion was enriched

References – incorrect format of journal references see: https://www.mdpi.com/journal/foods/instructions#references

Authors: the bibliography was modified according to the journal format

Reviewer 4 Report

Comments and Suggestions for Authors

You describe a fast and efficient method for the quantification of nine mycotoxins in beer using freeze-drying, reconstitution and extraction of the resulting powder followed by capture of the target mycotoxins by antibodies immobilized in a chromatographic column and finally quantification by ESI-MS. The method seems to have the potential to be used maybe not in breweries themselves but certainly by national food safety authorities.
Parts of your manuscript read not like a manuscript intended for publication but rather like a first internal draft. Instances where this was especially pronounced are mentioned below in the comments to specific lines in your manuscript.
There are also experimental details missing in your description of the applied methods (see below), please add those so an inclined reader may reproduce your results.
The reported mass per charge ratios of the precursor ions (Table 3) are approximately 0.5 m/z above the molecular weight of the uncharged compounds. ESI in positive mode should produce [M+H]+ ions with a m/z ratio equal to the molecular weight plus one. Please check your calibration and/or explain the deviation.

Comments pertaining to specific lines in the manuscript:

40f, “no biochemical significance for fungal development”: Secondary metabolites are often considered as biological weapons that hamper the competition's growth. Please reformulate the sentence to avoid giving the reader the impression that these compounds are useless to the fungi producing them.

66, “barley is the primary ingredient”: Other grains are also used for the production of beer, e.g. wheat.

95, “1,250 µg/kg”: This higher value refers specifically to "Unprocessed oat grains with inedible husk" i.e. the husks have to be removed before selling any derived product as a food.
The relevant values are from 10 ("Baby food and processed cereal-based food for infants and young children" up to, and of special relevance to your topic, 200 µg/kg ("Unprocessed malting barley grains"). The EC grants 50 µg/kg "extra" just for the designation as "malting" for barley grains.

105, “AOFZDT2TM column“: Please provide an actual product name that also appears in the catalogue of VICAM or describe the column chemistry in detail.

129-139: These sentences read like a sales text.

168-174: These sentences may be more appropriately placed in the conclusion section.

179f: What cut-off did you apply for the conductivity of your ultrapure water?

182: Please provide the technical specifications of the used filter papers (pore/mesh size, with or without binder, diameter, …).

184, “AOFZDT2TM“: This name does not appear in the catalogue of VICAM. Please provide the actual product name. Myco6in1+?
What are the dimensions of this column (particle size, inner diameter, length)?

186: Is "distilled water" different from the ultrapure water described above? If so, please specify (type of distiller, number of distillations, final conductivity, ...).

194: Please provide the details of the freeze-drying process. At least describe how you froze the samples and which temperatures (sample and trap) were used. The obtained dry weights should also be reported (in the SI, perhaps).

198: Beer is miscible with PBS. How did you extract beer with another aqueous solution?

199, “60 min”: Please provide the shacking frequency, the shaking style (with the respective parameters, e.g. orbit for circular shakers, used vessels) and the temperature during the shaking step.

199: At which temperature did you centrifuge?

200: Please specify which filter you used and how you did filtrate (by gravity in a funnel, using vacuum with the filter supported on a glass frit, ...).

206: At which temperature were the columns loaded?

207-209: Did you load both extracts on the same column for one HPLC-MS/MS run?
How was the recovery if only the aqueous or the alcoholic fraction were applied to the column? Please do explain why both fractions of the extract were needed.

211f: How did you check the elution of bound toxins for completeness?

214: What was in the solution used for sample reconstitution?

220, “500 µg/L”: Why did you chose a level 100-times higher than the other spikes?

227, “99:1”: Are the mixtures you present to be understood as volume fractions? Please clarify, e.g. 99:1 (v+v).

229f: How often did you prepare this mix? The preparation of the standard mix should be captured in the repeatability value.

233, “dissolving”: Do you mean "diluting" appropriate amounts of the starting mix solution?

243, Table 1: What were the nebulizing and heating gases?
Crytical parameters are missing: Please add the voltages.
The abbreviation "DL" is not defined. Do you mean "desolvation line"?

247: Please add a reference to Table 3.

249: At which temperature did the chromatographic separation take place?

251: Please clearly define mobile phases A and B.

252: “0.1% acetic acid”: Are those %(v/v)?

254, Table 2: Is this a step gradient or a linear one? Please clarify.
There is some confusion in your table: "Inject", "B Conc."
Please refine to a two columns table (Time [with unit] and %B) and define %A = 100% - %B.

263: How did you define and measure "signal" and "noise"?

268, “blank samples”: How did you manufacture your blanks? Ideally, blank samples undergo the complete sample preparation procedure.

277, “mycotoxins as model matrix”: The mycotoxins are the analytes, the rest of the beer forms the matrix.

290: “protonsified molecular ions“: Please use the standard term "protonated" (pseudomolecular ions).

296, Table 3: Please specify which of the two fragment ions was used for quantification and verification, respectively.
HT-2, precursor m/z of 442: What modification of HT-2 (MW: 424.48 Da) is responsible for the observed higher m/z ratio?
DON, fragment ion with higher m/z: What happened here? The addition of 9 m/z is curious.

329, “the fourth level (n=4) of each calibration curve (table 5)“: Please add these concentrations to Table 5.
Please do also report the total recovery (from beer to detector, including the freeze-drying and extraction steps) for toxin concentrations at or slightly below the legal limits.

330, “retention time”: This would be a good point to introduce the abbreviation "RT", which is so far undefined in your manuscript.

332, Tab. 4: Why do you report your values as ppb (i.e. mass based, m/m) instead of the more usual µg/L (mass concentration, m/v) that is also used in the regulatory documents?
Why is the linear range for the Aflatoxins so small?
It is not likely to have a higher variance on technical replicates (Repeatability, same experiment done again with the same samples, equipment and by the same people) than on samples run on different days by different people using different instrumentation (Reproducibility, same method applied to different samples by different people).
Please use either “Tab.” or “Table” consistently.

337, Table 5: The total recovery (from beer to detector readout) would be the interesting parameter. As you spiked beer with mycotoxin standards you should have the data already, please do report the overall recovery (at the relevant concentration levels, i.e. close to the legal limits).

341-345: These sentences may be better placed as figure legends.

347. Fig. 1a: What does "SCAN" mean here?
Please provide better figures: larger axis legends, clear description of shown curves (The internal sample name does not suffice.) and probably no gridlines (though the last one is a matter of taste).

347, Fig. 1b: Please add a note that the scale of the ordinate is different between Fig. 1a and 1b.

353, Fig. 2: This figure could be moved to the SI.
Please add (also in the SI) a figure showing the chromatograms of just the standards.

362, “the reduced sample volume minimizes potential interferences“: As the interferents present in the original beer are concentrated as well, please add an explanation why the interference is reduced.

Author Response

Dear Review4,

Thank you for your suggestions on our manuscript: “ An Innovative Analytical Approach for Multi-Mycotoxin Detection in Craft Beer using Freeze-Dried Samples, IAC column and HPLC/ESI-MS/MS”. The manuscript was modified according to your suggestions. The corrections or specific answers are listed below point by point.

You describe a fast and efficient method for the quantification of nine mycotoxins in beer using freeze-drying, reconstitution and extraction of the resulting powder followed by capture of the target mycotoxins by antibodies immobilized in a chromatographic column and finally quantification by ESI-MS. The method seems to have the potential to be used maybe not in breweries themselves but certainly by national food safety authorities.

Parts of your manuscript read not like a manuscript intended for publication but rather like a first internal draft. Instances where this was especially pronounced are mentioned below in the comments to specific lines in your manuscript.

There are also experimental details missing in your description of the applied methods (see below), please add those so an inclined reader may reproduce your results.

The reported mass per charge ratios of the precursor ions (Table 3) are approximately 0.5 m/z above the molecular weight of the uncharged compounds. ESI in positive mode should produce [M+H]+ ions with a m/z ratio equal to the molecular weight plus one. Please check your calibration and/or explain the deviation.

Comments pertaining to specific lines in the manuscript:

40f, “no biochemical significance for fungal development”: Secondary metabolites are often considered as biological weapons that hamper the competition's growth. Please reformulate the sentence to avoid giving the reader the impression that these compounds are useless to the fungi producing them.

Author response: the sentence “The production of secondary metabolites by filamentous fungi, which have no biochemical significance for fungal development, produces naturally occurring compounds with a low molecular weight known as mycotoxins” was changed to “The production of secondary metabolites by filamentous fungi results in naturally occurring, low-molecular-weight compounds known as mycotoxins, which, while not directly essential for fungal development, can play important roles as biological weapons that hamper the competition's growth”

66, “barley is the primary ingredient”: Other grains are also used for the production of beer, e.g. wheat.

Author reponse: That's true, but barley malt is by far the most widely used ingredient in beer production.

95, “1,250 µg/kg”: This higher value refers specifically to "Unprocessed oat grains with inedible husk" i.e. the husks have to be removed before selling any derived product as a food. The relevant values are from 10 ("Baby food and processed cereal-based food for infants and young children" up to, and of special relevance to your topic, 200 µg/kg ("Unprocessed malting barley grains"). The EC grants 50 µg/kg "extra" just for the designation as "malting" for barley grains.

Author response: According to the reviewer’s suggestion the sentence “The limits vary greatly depending on the type of raw material, with values ranging from 50 µg/kg to 1,250 µg/kg.” Was changed to “The limits vary greatly depending on the type of raw material, with values ​​ranging from 10 µg/kg (Baby food and processed cereal-based food for infants and young children) to 200 µg/kg (Unprocessed malting barley grains)”

105, “AOFZDT2TM column“: Please provide an actual product name that also appears in the catalogue of VICAM or describe the column chemistry in detail.

Author response:  According to the reviewer’s suggestion the name “AOFZDT2TM column“ was changed to “Myco6in1+ columns”

129-139: These sentences read like a sales text.

Author response: According to the reviewer’s suggestion these sentences were changed

168-174: These sentences may be more appropriately placed in the conclusion section.

Author response: According to the reviewer’s suggestion these sentences were muved

179f: What cut-off did you apply for the conductivity of your ultrapure water?

Author reponse: Our ultrapure water generator delivers water with a conductivity of just 0.058 µS/cm, ensuring the highest level of purity for your applications.

182: Please provide the technical specifications of the used filter papers (pore/mesh size, with or without binder, diameter, …).

Author response: Whatman® glass microfiber grade GF/C filter discs 1.2 μm pore size white, binder free, 0.26 mm thick, 100 ea, 47 mm diam

184, “AOFZDT2TM“: This name does not appear in the catalogue of VICAM. Please provide the actual product name. Myco6in1+?

Author response:  According to the reviewer’s suggestion the name “AOFZDT2TM column“ was changed to “Myco6in1+ columns”

What are the dimensions of this column (particle size, inner diameter, length)?

Author response: Column size of 6 mL

186: Is "distilled water" different from the ultrapure water described above? If so, please specify (type of distiller, number of distillations, final conductivity, ...).

Author response: This was an error; ultrapure water was used. The text was modified

194: Please provide the details of the freeze-drying process. At least describe how you froze the samples and which temperatures (sample and trap) were used. The obtained dry weights should also be reported (in the SI, perhaps).

Author response: Set value of main drying time 1 h vacuum -20°C 1,0 mbar, Set value of final drying time until completely drying, vacuum -53°C 0,025 mbar

198: Beer is miscible with PBS. How did you extract beer with another aqueous solution?

Author response: Yes, beer is miscible with PBS. PBS is an aqueous solution

199, “60 min”: Please provide the shacking frequency, the shaking style (with the respective parameters, e.g. orbit for circular shakers, used vessels) and the temperature during the shaking step.

Author response: an orbital incubator Stuart SI500 at 150 rpm, 25°C and 60 min.

199: At which temperature did you centrifuge?

Author response: 25°C

200: Please specify which filter you used and how you did filtrate (by gravity in a funnel, using vacuum with the filter supported on a glass frit, ...).

Author response: Whatman® glass microfiber grade GF/C filter discs 1.2 μm pore size white, binder free, 0.26 mm thick, 100 ea, 47 mm diam206: At which temperature were the columns loaded?

Author response: 25°C

207-209: Did you load both extracts on the same column for one HPLC-MS/MS run?

Author response: Yes

How was the recovery if only the aqueous or the alcoholic fraction were applied to the column? Please do explain why both fractions of the extract were needed.

Author response: A two-step extraction method, employing phosphate buffered saline (PBS) followed by methanol/water, was used to maximize the co-extraction of all investigated mycotoxins

211f: How did you check the elution of bound toxins for completeness?

Author response: Table 5 report the recovery %

214: What was in the solution used for sample reconstitution?

Author response: the samples were reconstituted with mobile phase (methanol/water 40:60, containing 1 mM ammonium acetate and 0.1% acetic acid)

220, “500 µg/L”: Why did you chose a level 100-times higher than the other spikes?

Author response: the 500 µg/L concentration was selected to balance providing a sufficiently high calibration point, avoiding potential issues with detector linearity and matrix effects, and maintaining practical feasibility. It effectively addressed the needs of our analysis without requiring an unnecessarily high and potentially problematic concentration

227, “99:1”: Are the mixtures you present to be understood as volume fractions? Please clarify, e.g. 99:1 (v+v).

Author response: 99:1 (v:v) was added

229f: How often did you prepare this mix? The preparation of the standard mix should be captured in the repeatability value.

Author response: the mix was prepared DON 1000 µg/L, HT-2 1000 µg/L, AFB1 5 µg/L, AFB2 5 µg/L, AFG1 5 µg/L, AFG2 5 µg/L, OTA 10 µg/L, FB1 500 µg/L, FB2 500 µg/L

233, “dissolving”: Do you mean "diluting" appropriate amounts of the starting mix solution?

Author response: changed

243, Table 1: What were the nebulizing and heating gases?

Author response: the nebulizing gas was Nitrogen and heating gas was Air

Crytical parameters are missing: Please add the voltages.

Author response: the voltage values are showed in the table 3 for Q1 e Q3

The abbreviation "DL" is not defined. Do you mean "desolvation line"?

Author response: Yes, the definition was added in the table as a footnote

247: Please add a reference to Table 3.

Author response: added

249: At which temperature did the chromatographic separation take place?

Author response: the chromatographic separation was taken a 25°C

251: Please clearly define mobile phases A and B.

Author response: added

252: “0.1% acetic acid”: Are those %(v/v)?

Author response: added

254, Table 2: Is this a step gradient or a linear one? Please clarify. There is some confusion in your table: "Inject", "B Conc." Please refine to a two columns table (Time [with unit] and %B) and define %A = 100% - %B.

Author response: added

263: How did you define and measure "signal" and "noise"?

The LOQ can be determined using various approaches, depending on whether the procedure is instrumental or non-instrumental, with alternative methods also being acceptable. One common approach is based on the standard deviation of the response and the slope of the calibration curve, expressed as:

LOQ = 10σ/S

where σ is the standard deviation of the response (signal), and S is the slope of the calibration curve (noise). The slope can be estimated from the analyte's calibration curve, which should be studied using samples containing the analyte at the QL range. The standard deviation can be derived from the residual standard deviation of the regression line or the standard deviation of y-intercepts from regression lines

268, “blank samples”: How did you manufacture your blanks? Ideally, blank samples undergo the complete sample preparation procedure.

Author response: in this method, we consider as a blank sample a beer that was analysed without the spik with the defined amount of mycotoxins standard. This beer was analysed before each analysis and then considered as blank.

277, “mycotoxins as model matrix”: The mycotoxins are the analytes, the rest of the beer forms the matrix.

Author response: yes

290: “protonsified molecular ions“: Please use the standard term "protonated" (pseudomolecular ions).

Author response: it was substituted

296, Table 3: Please specify which of the two fragment ions was used for quantification and verification, respectively.

Author response: The precursor ions and the two most intense product ions for each analyte were measured for quantification and identification (confirmation) respectively.

HT-2, precursor m/z of 442: What modification of HT-2 (MW: 424.48 Da) is responsible for the observed higher m/z ratio?

Author response: it was a mistake in the numbers. Now it has been modified and corrected

DON, fragment ion with higher m/z: What happened here? The addition of 9 m/z is curious.

Authors: it was a mistake in the number. Now it has been modified and corrected

329, “the fourth level (n=4) of each calibration curve (table 5)“: Please add these concentrations to Table 5.

Author response: the levels were added

Please do also report the total recovery (from beer to detector, including the freeze-drying and extraction steps) for toxin concentrations at or slightly below the legal limits.

Author response: In table 5 it was reported the value of recovery for each mycotoxins in the spiked beer

330, “retention time”: This would be a good point to introduce the abbreviation "RT", which is so far undefined in your manuscript.

Author response: added

332, Tab. 4: Why do you report your values as ppb (i.e. mass based, m/m) instead of the more usual µg/L (mass concentration, m/v) that is also used in the regulatory documents?

Authors: it was changed

Why is the linear range for the Aflatoxins so small?

It is not likely to have a higher variance on technical replicates (Repeatability, same experiment done again with the same samples, equipment and by the same people) than on samples run on different days by different people using different instrumentation (Reproducibility, same method applied to different samples by different people).

Author response: The linear range for Aflatoxins was chosen to be relatively small because the legal limit for Aflatoxin contamination is very low. Our primary focus was to ensure high accuracy and precision within the critical range close to this legal threshold, which is the most relevant for real-world applications. Additionally, working within this narrower range allowed us to optimize the method's sensitivity and performance where it matters most, ensuring compliance with regulations. As for the variance, while it might seem counterintuitive, the difference in repeatability and reproducibility variance can occur due to the inherent challenges of inter-day and inter-operator variability, particularly with highly sensitive measurements like those for Aflatoxins. However, the overall performance of the method remains within acceptable limits for both repeatability and reproducibility as specified by international standards.

Please use either “Tab.” or “Table” consistently.

Author response: modified

337, Table 5: The total recovery (from beer to detector readout) would be the interesting parameter. As you spiked beer with mycotoxin standards you should have the data already, please do report the overall recovery (at the relevant concentration levels, i.e. close to the legal limits).

Author response:  in this table it was reported the value of recovery for each mycotoxins in the spiked beer

341-345: These sentences may be better placed as figure legends.

Author response: moved in the legends

  1. Fig. 1a: What does "SCAN" mean here?

Author response: Scan means scansion of the Mass profile

Please provide better figures: larger axis legends, clear description of shown curves (The internal sample name does not suffice.) and probably no gridlines (though the last one is a matter of taste).

Author response:  the figures were changed with figures of high resolution that was taken by instrument software

347, Fig. 1b: Please add a note that the scale of the ordinate is different between Fig. 1a and 1b.

Author response:  the scale of ordinate was different because in the beer undoped the level of mycotoxins was lowest and so the scale was modified so to show in a good way the peaks.

353, Fig. 2: This figure could be moved to the SI. Please add (also in the SI) a figure showing the chromatograms of just the standards.

Author response: The figures were modified, but their positions remain unchanged, contrary to the other reviewer's recommendation

362, “the reduced sample volume minimizes potential interferences “: As the interferents present in the original beer are concentrated as well, please add an explanation why the interference is reduced.

Author response: this sentence was rewritten

Reviewer 5 Report

Comments and Suggestions for Authors

This study presents an innovative analytical protocol using liquid chromatography/electrospray ionization tandem mass spectrometry (HPLC/ESI MS/MS) for the simultaneous qualitative and quantitative analysis of nine mycotoxins, including aflatoxins (AFB1, AFB2, AFG1, AFG2), Ochratoxin A (OTA), Fumonisins (FB1, FB2), Deoxynivalenol (DON), and HT-2. There are several issues in this paper.

(1) line 105 Add references

(2) lines 108-125 Reduce to one sentence

(3) The introduction is too long. Please clarify the novelty and contribution of this study more explicitly in Introduction.

(4) 2.1.1 Samples Add the types of beer

(5) ppb Please use standard units

(6) Fig. 1 and 2. Redraw with the professional drawing software. 

(7) Add a comparative discussion with traditional methods. 

Author Response

Review 5

Thank you for your suggestions on our manuscript: “An Innovative Analytical Approach for Multi-Mycotoxin Detection in Craft Beer using Freeze-Dried Samples, IAC column and HPLC/ESI-MS/MS”. The manuscript was modified according to your suggestions. The corrections or specific answers are listed below point by point.

Review: This study presents an innovative analytical protocol using liquid chromatography/electrospray ionization tandem mass spectrometry (HPLC/ESI MS/MS) for the simultaneous qualitative and quantitative analysis of nine mycotoxins, including aflatoxins (AFB1, AFB2, AFG1, AFG2), Ochratoxin A (OTA), Fumonisins (FB1, FB2), Deoxynivalenol (DON), and HT-2. There are several issues in this paper.

  • line 105 Add references

Authors: done

  • lines 108-125 Reduce to one sentence

Authors: the sentences were rewritten

  • The introduction is too long. Please clarify the novelty and contribution of this study more explicitly in the Introduction.

Authors: the introduction was summarized and rewritten

  • 1.1 Samples Add the types of beer

Authors: the analysed samples were three lager beers Bottom-fermented, Pilsner style

(5) ppb Please use standard units

Authors: done

(6) Fig. 1 and 2. Redraw with professional drawing software. 

 Authors: The figures were replaced

(7)  Add a comparative discussion with traditional methods.

Authors: A table (table 7) for the comparison was added

Round 2

Reviewer 2 Report

Comments and Suggestions for Authors

The revised document significantly enhances the understanding of mycotoxin detection in craft beer, addressing essential consumer safety and regulatory compliance issues. It delves into the complexities of mycotoxin contamination in brewing, highlighting the necessity for effective monitoring due to the presence of various fungal species in raw materials like barley.

The innovative analytical protocol using HPLC/ESI-MS/MS improves detection capabilities and streamlines the sample preparation process, which is crucial for achieving accurate results. By validating this method with craft beers from Calabria and adhering to European Commission regulations, the authors demonstrate their commitment to advancing food safety standards in the brewing industry.

This methodology offers several enhancements to food safety: it provides comprehensive detection of multiple mycotoxins, including aflatoxins and ochratoxin A; improves sample preparation through immunoaffinity columns for better specificity and sensitivity; ensures regulatory compliance, thereby protecting consumers from potential health risks; and specifically addresses mycotoxin levels in craft beers, acknowledging the unique challenges faced by smaller breweries.

Overall, the proposed methodology represents a significant advancement in food safety by filling knowledge gaps and providing a robust tool for quality control. Thus, it underscores the importance of stringent regulatory measures in preventing mycotoxin-related health risks.

Author Response

Thank you for your detailed and insightful comment. We are pleased that the document has improved the understanding of mycotoxin detection in craft beer and highlights key aspects of consumer safety and regulatory compliance.

We appreciate your recognition of the methodological approach, particularly regarding detecting multiple mycotoxins, using immunoaffinity columns for greater selectivity, and consumer protection. Our goal is to provide industry professionals with a reliable quality control system, paying special attention to the unique challenges faced by small breweries.

Reviewer 3 Report

Comments and Suggestions for Authors

Apart from some minor errors (see below) there is one thing that should be addressed by the authors. There is positive bias in results for the freeze-dried samples. This occurrence should be investigated, for example by preparing a set of calibration/spiked solution and subject them to the freeze-drying process in order to clarify whether the bias originates from the sample preparation procedure.

Please check following details:

Line 396 – reproducibility

Line 396 – rounding error for Aflatoxin B2

Author Response

Apart from some minor errors (see below) there is one thing that should be addressed by the authors. There is positive bias in results for the freeze-dried samples. This occurrence should be investigated, for example by preparing a set of calibration/spiked solution and subject them to the freeze-drying process in order to clarify whether the bias originates from the sample preparation procedure.

Author: We acknowledge the reviewer's observation regarding the positive bias in the results for freeze-dried samples. Our findings indicate that the bias originates from the sample preparation procedure, likely due to the concentration of analytes during the freeze-drying process. This concentration effect leads to higher apparent levels of mycotoxins in the freeze-dried samples compared to the liquid samples.

Please check following details:

Line 396 – reproducibility

Author: After careful review, we have not been able to identify the error in the line you indicated.

Line 396 – rounding error for Aflatoxin B2

Author: After careful review, we have not been able to identify the error in the line you indicated.

Reviewer 5 Report

Comments and Suggestions for Authors

Accept in present form

Author Response

Thank you for your comment. We're pleased the document has improved understanding of mycotoxin detection in craft beer, highlighting key aspects of consumer safety and regulatory compliance